# Effects of Livestock Pressure and Vegetation Cover on the Spatial and Temporal Structure of Soil Microarthropod Communities in Iberian Rangelands

**Carlos Lozano Fondón** [1,*] , **Jesús Barrena González** [2] , **Manuel Pulido Fernández** [2] ,
**Sara Remelli** [1] , **Javier Lozano-Parra** [3] **and Cristina Menta** [1]

[1] Department of Chemistry, Life Sciences and Environmental Sustainability, University of Parma,
   Viale delle Scienze 11/A, 43124 Parma, Italy; sara.remelli@unipr.it (S.R.); cristina.menta@unipr.it (C.M.)
[2] GeoEnvironmental Research Group, University of Extremadura, Avda. Universidad s/n, 10071 Cáceres,
   Spain; jesusbarrena@unex.es (J.B.G.); mapulidof@unex.es (M.P.F.)
[3] Instituto de Geografía, Pontificia Universidad Católica de Chile, Avda. Vicuña Mackenna 4860,
   Santiago de Chile 7820436, Chile; jlozano@uc.cl
*   Correspondence: lzncls@unife.it

**Abstract:** Forests, including their soils, play an important role since they represent a large reservoir of biodiversity. Current studies show that the diversity of soil fauna provides multiple ecosystem functions and services across biomes. However, anthropogenic practices often pose a threat to soil fauna because of changes in land use and soil mismanagement. In these terms, rangelands in the southwest of Spain present several problems of soil degradation related to livestock activity and soil erosion, the intensity of which compromises the soil fauna's functions in the ecosystem. Therefore, the aim of this study is to evaluate the response of community metrics and the spatial distribution of soil microarthropods to livestock activity and vegetation in such ecosystems. A photo interpretation analysis of an experimental catchment used as a study area was developed to identify and classify the intensity of livestock pressure. A total of 150 soil samples were collected throughout 2018. Soil biological ($CO_2$ efflux) and physical-chemical parameters (pH, bulk density, organic matter, and water contents), and such meteorological variables as precipitation, temperature, and evapotranspiration were considered as variables affecting the composition of microarthropod communities in terms of taxa diversity, abundances, and their adaptation to soil environment (evaluated by QBS-ar index). Results showed higher abundance of microarthropods and higher adaptation to soil environment outside the influence of trees rather than beneath tree canopies. Moreover, the classification of livestock pressure revealed by the photo interpretation analysis showed low correlations with community structure, as well as with the occurrence of well-adapted microarthropod groups that were found less frequently in areas with evidence of intense livestock activity. Furthermore, abundances and adaptations followed different spatial patterns. Due to future climate changes and increasing anthropogenic pressure, it is necessary to continue the study of soil fauna communities to determine their degree of sensitivity to such changes.

**Keywords:** arthropod-based soil quality; community structure; environmental filtering; remote sensed image analysis; spatial heterogeneity; morphological adaptation

## 1. Introduction

Soils are one of the most important reservoirs of biodiversity in the world [1–6]. Today, problems such as climate change, an increasing of human population, changes in land use, and land abandonment compromise the diversity and functions of soil biota [7,8] and, subsequently, the provision of ecosystem

services [2,4,8–10] by the soil complex. Moreover, soil functions and healthy soil communities are closely correlated, and, together, they are essential for safe and sustainable food production [9], and they also maintain ecosystem stability and resilience [8]. Furthermore, the diversity of the soil community is often used to provide soil quality indicators, such as the composition and abundance of microarthropod communities [11,12]. Indeed, it is widely accepted that soil microarthropods are very sensitive to disturbances because of their adaptation to a soil environment [12,13]. Although the contribution of microarthropods to the total amount of energy fluxes and biogeochemical transformations occurring in the soil is relatively low [14], they are a key component in enhancing the resilience and resistance of the soil food web by supporting structural stability [15] since they link microorganisms to macrofauna in the context of an interconnected network [16]. Such close correlations between bacterial and fungal channels to mesofauna [9] also determine top-down and bottom-up forces that modify the structure of the entire community and, therefore, the efflux of $CO_2$ produced by the soil food web during its metabolic activity [3].

Many ecological functions have been attributed to soil microarthropod communities [17]. However, the functions they perform can be compromised via the reduction of biodiversity caused by disturbances in the soil environment [2]. Therefore, the loss of functional groups of microarthropods, such as detritivores, which are related to the soil carbon cycle could determine the interruption of several steps in the organic matter degradation chain [18,19]. In such a context, Mediterranean bioclimatic areas with semi-arid conditions, such as rangelands in the Southwestern Iberian Peninsula, are susceptible to this fact since they have been catalogued as ecosystems under risk due in future climate scenarios to land mismanagement and livestock intensification [20].

*Dehesas* are traditionally-managed rangelands commonly characterized by a two-layered vegetation structure: a savanna-like open tree layer (15–40 trees/ha) with an understorey pasture in the same land unit [21–24]. Moreover, pools of soil nutrients are frequently limited due to poor parent material and extremely arid conditions during the Mediterranean summer [21,24]. It is a system particularly subject to abandonment [22,25], soil degradation [22,24,26], and subsequent loss of soil biodiversity because of the increase of livestock density and the progressive abandonment of land by farmers. However, patches of vegetation are important for *dehesas* to maintain biodiversity associated with spatial heterogeneity [27]. In this context, trees play an important role in regulating environmental features such as soil temperature [28,29] and moisture [23]; the modification of chemical characteristics, such as availability of nutrients [30], and the direct promotion of the development of detritivorous microarthropod communities via the reduction of sunlight availability and litter inputs [31]. Moreover, such habitat heterogeneity at multiple spatial scales [22,30] could represent areas for the conservation of biodiversity in farmlands, as indicated by Moreno et al. [27] in which the authors used the term "habitat condition" to refer to areas that sustain certain levels of aboveground biodiversity in rangelands [32]. We adapted this concept to our study area in trying to define combinations of environmental features and elements of the landscape (mostly in reference to vegetation and livestock pressure) that could also drive the spatial distribution, structure, abundances, and adaptation to soil of microarthropod communities. In order to clearly define such combinations of factors in this work, we used the term "soil habitat condition" (SHC) [32].

Thus, the central questions of our study are: do different intensities of livestock activity induce changes in soil microarthropod communities? Is the structure of a microarthropod community affected by niche environmental factors associated with the presence of the tree? Do the adaptations and abundances of microarthropods follow different spatial patterns? With regard to these questions, we defined mainly three aims: (i) determining changes on microarthropod communities associated to seasonality, proximity to trees, and intensity livestock pressure; (ii) identifying the most sensitive biological forms of the microarthropod group to livestock pressure; and (iii) exploring the spatial patterns of microarthropod abundances and the occurrence of morphological traits that indicate high adaptation to the soil environment.

## 2. Study Area

Research was conducted on a farmland with agro-silvo-pastoral land use located in the province of Cáceres, in the southwest of Spain, where an experimental catchment was delimited (Figure 1). The study area (151.6 ha) is representative of a traditionally-managed system, commonly known as a *dehesa*, which is dominated by several vegetation layers including scattered oak trees (*Quercus ilex* L.), a shrub layer (*Retama sphaerocarpa* L.), and a herbaceous layer composed of annual species (grasses such as *Vulpia bromoides* L. [Gray], *Bromus* sp., *Aira caryophyllea* L., and legumes such as *Ornithopus compressus* L., *Lathyrus angulatus* L., and several species of *Trifolium*) [21,23]. Climate (Table A1 in Appendix A) is typical of the Mediterranean area, with semi-arid conditions characterized by cold winters and a period of hydric stress during the summer. Mean annual precipitation is about 524.2 mm. Rainfall events are common in autumn and spring; however, dry seasons and longer dry periods are frequent. Mean annual temperatures oscillates from 14° to 16 °C.

Geomorphologically, the study area is in old erosion surfaces (Figure 1A), which are formed by schist and greywacke of the Precambrian age [26]. Soils are shallow with a thickness of usually less than 50 cm [24,26]; soil textures are sandy-loam in low-slope areas and silty-loam in areas with a higher slope. Soils reactions oscillate from 4.3 to 7.3, and they are poor in organic matter (mean values are about 3% in the A horizon) [26]. They are classified as Luvisols and Cambisols [33].

Farm management is conventional: livestock walk freely inside the farm, which means that livestock charges per hectare inside the study area are not equally distributed. Moreover, the presence of several "points of reunion," such as eating zones and water reservoirs, influence the frequency of trampling and grazing of surrounding areas close to them. In 2018, the livestock at the farm comprised 1200 sheep and goats (southeast area), 50 pigs (northwest area), 37 cows, and one bull (southwest and central areas of the farm).

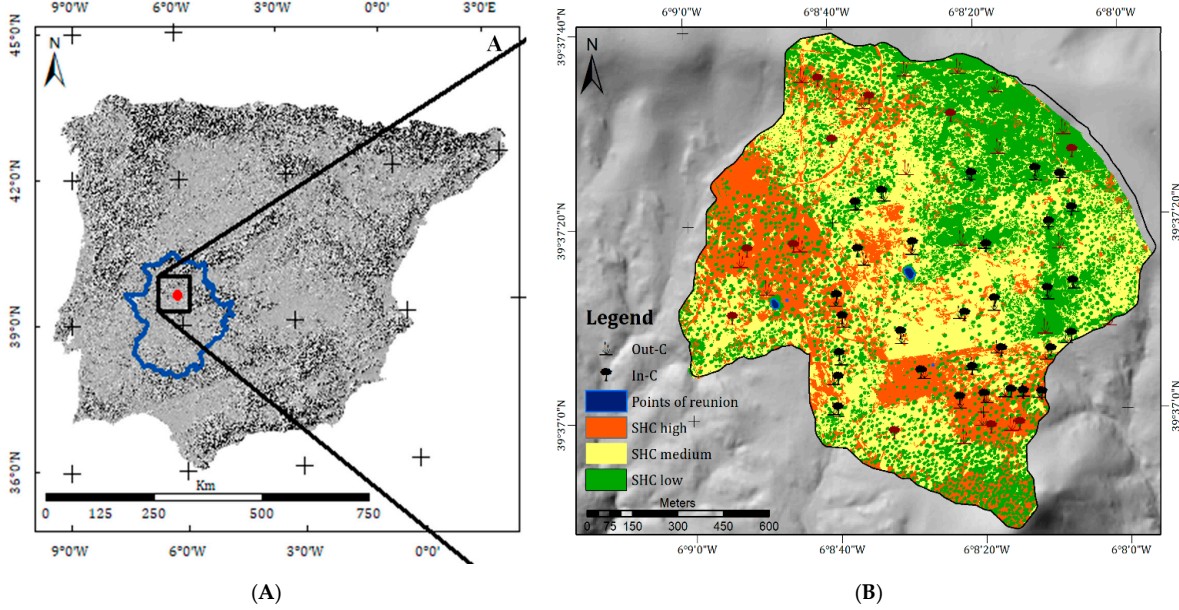

**(A)**    **(B)**

**Figure 1.** (**A**) Study area within the Extremadura region (blue boundaries) in the Iberian Peninsula and (**B**) the result of the object-based image analysis (OBIA) classification for the entire experimental catchment. Red indicates the ensemble of characteristics defining the $SHC_{high}$ area, yellow represents $SHC_{medium}$, and $SHC_{low}$ is indicated by green. Sampling point distribution across the SHC categories is also shown: points sampled in spring are black and points added in autumn are indicated by dark red. A tree symbol (in-C; black or dark red) indicates the geographic location of a sampling point beneath the tree canopy; a grass symbol (out-C; black or dark red) indicates the geographic position of sampling points outside the canopy.

## 3. Materials and Methods

### 3.1. Determination of the Intensity of Livestock Pressure

The study involved a description of the farm management by interpreting orthoimages (0.5 MP size) taken in 2016 by the Spanish National Information Center [34]. Parameters such as the density of the vegetation cover and the bare soil area were identified and related to livestock activity (mostly trampling and grazing) [30]. For the identification of zones with different grazing and trampling intensities, a supervised object-based image analysis (OBIA) classification [35,36] was used. The procedure was developed in the eCognition Developer 9 software (Trimble Germany Gmbh, Munich, Germany), avoiding the "salt and pepper effect" that occurs with pixel-oriented classifications [37,38].

Broadly, three categories were defined by OBIA based on the effects of livestock activity and the characteristics of the vegetation cover (Figure 1B). We then confirmed the field classifications: (1) $SHC_{low}$: characterized by a shrub-encroached herbaceous layer, typically 40%–70% *Retama sphaerocarpa* L. cover with a dense tree layer, absence of bare soil and no signs of livestock pressure (i.e., defecation, trampling, or grazed vegetation); (2) $SHC_{medium}$: herbaceous layer, mostly 10%–40% of *R. sphaerocarpa* L. cover with a sparse tree layer, <10% of bare soil, and slight indicators of livestock presence, and (3) $SHC_{high}$: herbaceous layer with a sparse tree layer but no shrub cover, 50% or more bare soil, and evident signs of livestock pressure. See Figure 2 for an example of the general characteristics of each SHC on the field.

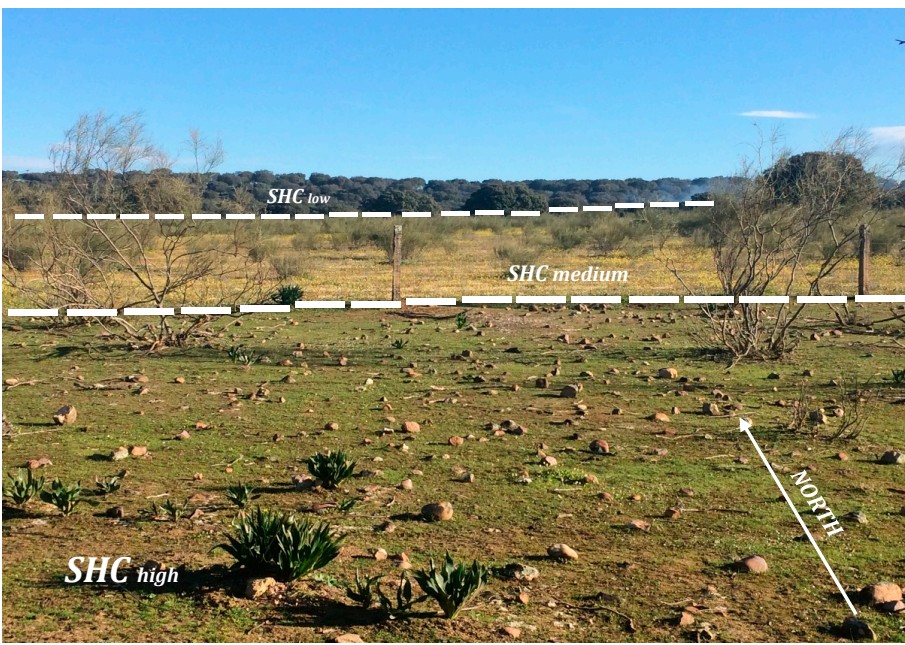

**Figure 2.** A picture taken in the study area showing the general characteristics of the environments classified as $SHC_{high}$, $SHC_{medium}$, and $SHC_{low}$.

### 3.2. Soil Sampling

Two sampling campaigns were carried out in 2018. A total of 60 points were sampled in spring (April), and 90 in the late fall (December). In both campaigns, points were equally distributed among the three SHC categories previously identified by OBIA (Figure 1B). Inside each SHC, half of the points were established beneath the oak canopies (in-C) on the northern cardinal point in relation to the stem. The other half were located outside the canopy (out-C), at least 8 m far away from every tree (stem) [18]. Both in-C and out-C points were established considering the presence of herbaceous vegetation cover and avoiding bare soil because it is widely accepted that low densities of soil fauna occur in absence of vegetation [13,39]. We made this choice to accurately compare the different SHCs because bare ground was only present in $SHC_{high}$.

For the spring sampling, 60 PVC cylinders (21 cm ø and 15 cm height) were embedded at each point the day before the beginning of the sampling campaign, as recommended by the LI-COR 8100A protocol of soil $CO_2$ efflux measurement [40]. The cylinders were installed up to 10 cm deep, leaving 5 cm of the total height free in order to carry out the measurement correctly. After the stabilization of the microbial soil community [40], sampling was conducted as follows at each point: three repeated measurements of soil $CO_2$ efflux using a LI-COR 8100A survey device were executed; then three measurements of soil moisture were taken in three points beside the cylinder using a TDR device and, finally, three undisturbed soil cores were collected using a steel cylinder at a known volume (100 $cm^3$) and a soil sample extractor. The soil volumes collected from inside the cylinders used for the soil $CO_2$ efflux measurement (approx. 3.5 $dm^3$) were taken to the laboratory, where microarthropod extraction was carried out. Once organisms were collected, soil samples were meshed at 2 mm ø. Then, six replicates were picked up to determine the pH and soil organic matter (SOM) content of each sample (3 for pH and 3 for SOM). Due to logistical reasons, measurements of soil $CO_2$ efflux, pH, water content, and bulk density could not be revealed in the autumn campaign. At this time, only SOM has been measured in the 90 points sampled.

In the laboratory, pH was determined by dissolving 1 g of soil in 3 g of $H_2O$ using a pH meter. SOM was revealed using the loss-on-ignition method. Soil cores inside the steel cylinders were used to calculate soil bulk density and gravimetric moisture by the wet-minus-dry weights of the samples in relation to the volume of the undisturbed soil cores (100 $cm^3$).

### 3.3. Analysis of the Microarthropod Communities

The analysis of the microarthropod community was based on the QBS-ar methodology [11,12]. The QBS-ar index (i.e., biological soil quality based on arthropods) evaluates the capacity of a soil to harbor animals that are sensitive to disturbances because of their morphological characteristics. Therefore, based on the number of well-adapted microarthropods to the soil environment at a given time, it is possible to make a judgment about the quality of the soil in a given area (i.e., the higher the number of such organisms, the higher the soil quality). The QBS-ar of a soil sample is calculated as the sum of the ecomorphological indices (EMIs) of each biological form. The EMI is a dimensionless score that varies between 1 and 20, and it evaluates the degree of adaptation of the morphological traits that soil animals share by evolutionary convergence. For more details on QBS-ar application, see [13,17].

In this study, soil microarthropods were extracted from the 150 samples (both spring and autumn) using Berlese-Tullgren funnels (2 mm mesh size) and conserved in 70% ethanol solutions. The extraction time was about eight days, depending on the humidity of samples. Then, the extracted microarthropods were observed under stereomicroscope (40×), counted, identified, and classified as indicated in Table A2 from Appendix A. Once analysis of the microarthropod community was completed and raw data were collected, taxa diversity was evaluated via the Shannon's index; a value of QBS-ar was associated with each soil sample, and the numbers of individuals per taxon for each soil sample were obtained.

### 3.4. Statistical Analyses

Three response variables were considered for this work: abundances, taxa diversity (defined by Shannon's index), and community adaptation to soil environment (defined by the QBS-ar index). A first screening of the data was carried out following the protocol proposed by Zuur et al. [41]. Three categorical predictors were considered: (i) the season in which samplings were accomplished; (ii) the location of each sampling point (in-C: beneath tree canopy, out-C: outside the canopy); and (iii) the SHC representing the intensity of livestock activity surrounding each sampling point.

Initially, a three-way ANOVA test was performed to evince statistical differences among categories. Abundances were log-transformed based on the range of the data and significance level was established at $p = 0.05$. A Tukey pairwise test was applied post hoc to highlight the significant differences between

pairs of categories. Once seasonal variation of community metrics was statistically confirmed, the rest of the analyses were carried considering seasons separately.

Prior to analysis, collinearity was tested with Pearson's correlation coefficient in order to eliminate variables with identical trends [41]. When Pearson's correlation was found to be higher than 0.4999, covariates were considered as collinear and, subsequently, one was excluded from the analysis [42]. Methods such as, non-metric multidimensional scaling (NMDS) and non-parametric permutational multivariate ANOVA (PERMANOVA) were chosen to study dissimilarities in microarthropod communities. NMDS based on Bray–Curtis distances was used to order the relationships among communities' compositions in a specified number of axes [42]. A stress level score of ≤0.2 was used to account for goodness of fit. PERMANOVA, also based on Bray–Curtis distances, was then used to study environmental variables causing dissimilarity in the community structure [43,44]. In order to identify the sensitivity of each biological form, NMDS was also applied on each *taxon* based on the Bray-Curtis dissimilarity index. In order to accomplish these aims, community matrices were split into (1) two log-transformed abundance matrices (60 soil samples (rows) × 27 taxa (columns) in spring; 90 soil samples (rows) × 27 taxa (columns) in autumn); and (2) two EMI value matrices (60 soil samples (rows) × 27 taxa (columns) in spring, and 90 soil samples (rows) × 27 taxa (columns) in autumn) representing the morphological adaptation of biological forms to the soil environment. Environmental factors were summarized in a matrix presenting soil parameters, categorical predictors (SHC and out-C/in-C locations), geospatial characteristics (UTM coordinates, slope, and altitude of each sampling point), and meteorological variables such as maximum, minimum and mean temperature of the sampling day; average of maximums, minimums and mean temperatures of the 20 days prior to the sampling day, effective precipitation of the sampling day, effective cumulative precipitation of the 20 days prior to the sampling day, evapotranspiration of the sampling day, average evapotranspiration of the 20 days prior to the sampling day, and average hydrological balance of the 20 days prior to the sampling day (22 columns in spring and 18 in autumn). A stepwise model selection based on the significance criterion was used to choose the best combination of variables explaining the variance of the data. These analyses were carried out with the "vegan" package [45] from RStudio.

Generalized additive models (GAMs) were applied to investigate the effects of spatial distribution of niche-environmental factors upon spatial distribution of QBS-ar and total log-transformed abundances. In order to model the dispersion of community metrics across the space, two protocols were executed to run the models: the first, a random effect on "pure" spatial coordinates was used in order to seek spatial dependence of the response variables; on the second, a random effect on ordination coordinates extracted from NMDS replaced the spatial coordinates in order to find the best descriptor of the community metrics variation [41]. Stepwise model selection was based on Akaike's information criterion (AIC) [46]. The R package "mgcv" [47] was used to perform this analysis.

Graphics were generated using the "ggplot2" [48] package from RStudio.

## 4. Results

### 4.1. Soil Parameters

Organic matter and water contents, bulk density and pH were measured in the three SHC in both in out-C and in-C locations (Table 1). Averages slightly differed based on SHC categorization. However, decreasing values of organic matter content were found from $SHC_{high}$ to $SHC_{low}$ in out-C locations; differences were less evident when pH, bulk density, and water content were compared. Soil pH was found to be acidic (5.61 to 5.96) inside the study area with no broad variations both in either out-C or in-C. Moreover, bulk density averages in out-C varied slightly around 1.5 g cm$^{-3}$. However, values of the soil parameters on in-C were smaller in the case of bulk density (≈1.2 g cm$^{-3}$), but far higher in organic matter (≈10% compared to ≈5% for in-C and out-C, respectively). Instead, water content and soil $CO_2$ efflux were the less-variable parameters considering both out-C and in-T locations (≈20% and ≈5 μmol m$^{-2}$ s$^{-1}$, respectively).

**Table 1.** Mean ± standard deviations of soil parameters. SHC indicates the "soil habitat condition" categories; In-C and Out-C indicate whether values were detected beneath the tree canopy or outside the canopy, respectively.

| | Units | $SHC_{high}$ | | $SHC_{medium}$ | | $SHC_{low}$ | |
|---|---|---|---|---|---|---|---|
| | | Out-C | In-C | Out-C | In-C | Out-C | In-C |
| **Bulk density** | g cm$^{-1}$ | 1.5 ± 0.1 | 1.2 ± 0.2 | 1.5 ± 0.1 | 1.2 ± 0.2 | 1.5 ± 0.2 | 1.2 ± 0.1 |
| **Organic matter** | % | 5.3 ± 2.5 | 10.3 ± 4.6 | 1.9 ± 1.8 | 9.1 ± 4.6 | 3.8 ± 1.4 | 9.5 ± 4.1 |
| **pH** | - | 5.8 ± 0.7 | 6.0 ± 0.8 | 5.6 ± 0.4 | 5.9 ± 0.4 | 5.8 ± 0.2 | 5.7 ± 0.7 |
| **Soil CO$_2$ efflux** | μmol m$^{-2}$ s$^{-1}$ | 4.6 ± 2.8 | 4.9 ± 2.7 | 4.5 ± 1.4 | 5.2 ± 1.9 | 5.2 ± 1.9 | 5.0 ± 2.2 |
| **Water content** | % | 18.4 ± 5.9 | 20.8 ± 8.3 | 22.2 ± 8.8 | 23.2 ± 8.0 | 21.8 ± 8.4 | 23.7 ± 10.5 |

*4.2. Seasonallity, Environmental Characteristics, and Community Structure*

Generally, 113,579 organisms belonging to 27 taxa were individually identified and counted. Collembola, Acari, larvae of dipterans, and larvae of coleopterans were the most frequent taxa (58%, 35%, 3%, and 1%, respectively) representing 97% of the total abundances. In detail, frequencies varied significantly from spring to autumn ($p < 0.01$), as evidenced by results of the three-way ANOVA showed in Table 2. In spring, collembolans represented 50% of total abundances, which increased to 62% in autumn. Mite populations were higher in spring (44%) than in autumn (32%), and larvae of dipterans and larvae of coleopterans maintained their populations varying from 2% to 3% in the case of dipterans, remaining at 1% in the case of coleopterans. Other taxa, such as coleopteran adults, hemipterans, pauropods, thysanopterans, and ants, were frequently found but their abundances were lesser. Moreover, diplopods were found only in spring, and isopods conversely only in autumn, as well as individuals belonging to Zygentoma taxon. However, the most important source of variation was the location of each sampling point (in-C or out-C), showing a strong influence on the response variables. Although SHC was not revealed as a significant source of variation by the ANOVA, some taxa, such as diplopods, were found only in $SHC_{low}$ when trees were present. Embiopterans were found in the three SHCs, both in out-C and in-C, during spring campaign, but only in in-C locations during autumn. Proturans and pseudoscorpions followed similar patterns, as they were found in the same categories, but pseudoscorpions lacked in autumn in-C-$SHC_{low}$. Such variations were also reflected by the QBS-ar and H' indices, the values of which differed not only according to seasonality, but also to the location of sampling points (F = 4.490 and F = 6.232, both significant, respectively) as showed in Figure 3 and Table 2. The highest mean value of QBS-ar was detected in out-C-$SHC_{medium}$, followed by out-C-$SHC_{high}$, both in autumn. Generally, higher values of QBS-ar were found in out-C. Instead, mean values of H' were closer to 1 in in-C locations in spring, but not in autumn.

**Table 2.** Three-way ANOVA on log-transformed abundances, QBS-ar and Shannon's index (H'). Asterisks indicate levels of significance (*) = $p < 0.05$; (**) = $p < 0.01$; (***) = $p < 0.001$. "Location" indicates whether the sampling point was located beneath tree canopies (in-C) and outside the canopy (out-C).

| | Ln Abundances | | QBS-ar | | H' | |
|---|---|---|---|---|---|---|
| **Suorce of Variation** | F Test | *p*-Value | F Test | *p*-Value | F Test | *p*-Value |
| Livestock pressure | 2.911 | 0.058 | 2.451 | 0.090 | 1.532 | 0.220 |
| Location | 14.655 | <0.001 *** | 17.464 | <0.001 *** | 0.913 | 0.341 |
| Season | 7.644 | 0.007 ** | 2.057 | 0.154 | 0.007 | 0.932 |
| Livestock pressure × Location | 1.233 | 0.295 | 1.059 | 0.350 | 1.209 | 0.302 |
| Livestock pressure × Season | 1.355 | 0.262 | 0.511 | 0.601 | 2.247 | 0.110 |
| Location × Season | 0.015 | 0.902 | 4.490 | 0.036 * | 6.232 | 0.014 * |
| Livestock pressure × Location × Season | 1.718 | 0.184 | 1.581 | 0.210 | 0.634 | 0.532 |

See details in Appendix A: results of the post hoc Tukey test are shown in Table A3, mean values of QBS-ar and H' are in Tables A4 and A5, and total abundances are in Tables A6 and A7.

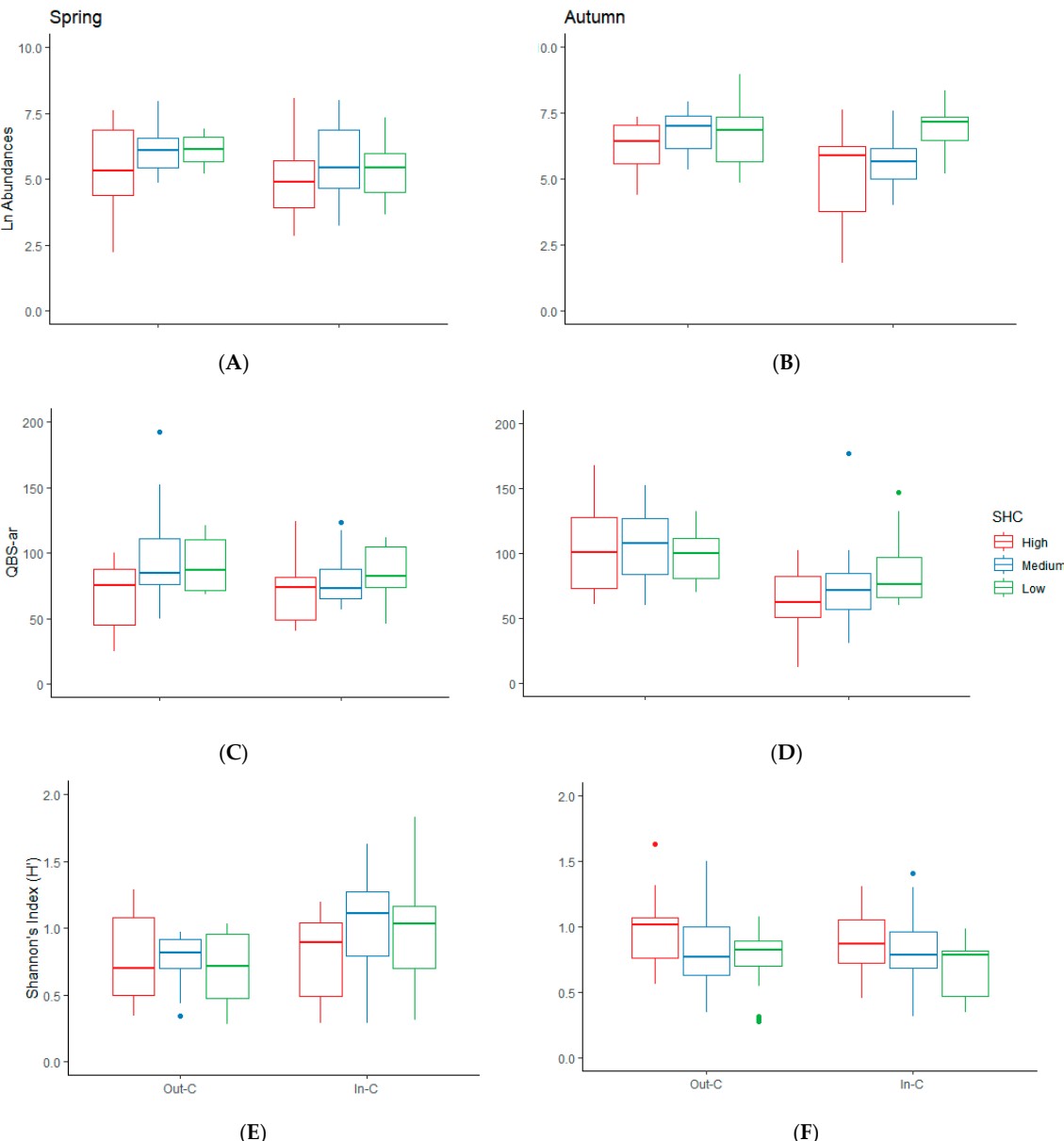

**Figure 3.** Boxplots showing the distribution of data for each response variable by soil habitat condition (SHC) and location factor (outside the canopy = Out-C or beneath tree canopy = In-C) in both seasons. (**A**,**B**) plots show log-transformed abundances; (**C**,**D**) plots show QBS-ar values; (**E**,**F**) plots show H′ index values.

Results of NMDS and PERMANOVA are reported in Figure 4 and Table 3, respectively. The ordination of sampling points based on community abundances showed stress values below the 0.15 threshold (Figure 4A,B). However, the occurrence of high EMI scores did not show convergence, as evidenced by stress values above 0.20 (Figure 4C,D). The scaling of community abundances was characterized by high overlapping based on categorical predictors (locations and SHC). Nevertheless, more dissimilarity among communities was attributed to locations ($p < 0.05$ and $p < 0.001$ in spring and autumn, respectively) than to SHC, which was significant in autumn ($p < 0.05$) but not significant at all in spring (Table 3). Representation of significant taxa abundances in spring was lower than in autumn, as shown in Figure 4A,B. In spring, such groups as Acari and Coleoptera (adults and larvae) avoided points where values of organic matter and pH were higher, but they were also positively related to high bulk density. Otherwise, Collembola, Thysanoptera, larvae of Diptera, and larvae of Coleoptera were positively related to points where soil water content was higher. In autumn

(Figure 4B), Collembola, Chilopoda, and Pauropoda avoided points where SOM was higher. Moreover, these groups were significantly related to out-C-SHC$_{medium}$ and out-C-SHC$_{low}$ in autumn. Several environmental variables were significant in both seasons, but after the model selection performed on PERMANOVA, only water content ($p < 0.01$), pH ($p < 0.05$), and soil $CO_2$ efflux ($p < 0.05$) were deemed significant ($R^2 = 17.4\%$). In the autumn model, environmental variables, such as slope ($p < 0.05$), SOM ($p < 0.01$), effective precipitation ($p < 0.05$), and mean temperatures of the 20 days prior to sampling day ($p < 0.05$), explained a wider percent of the total variance of PERMANOVA ($R^2 = 22.1\%$) as compared to the spring model. It is noteworthy that the location of sampling points was found significant in both abundance models ($p < 0.05$ and $p < 0.001$ in spring and autumn, respectively).

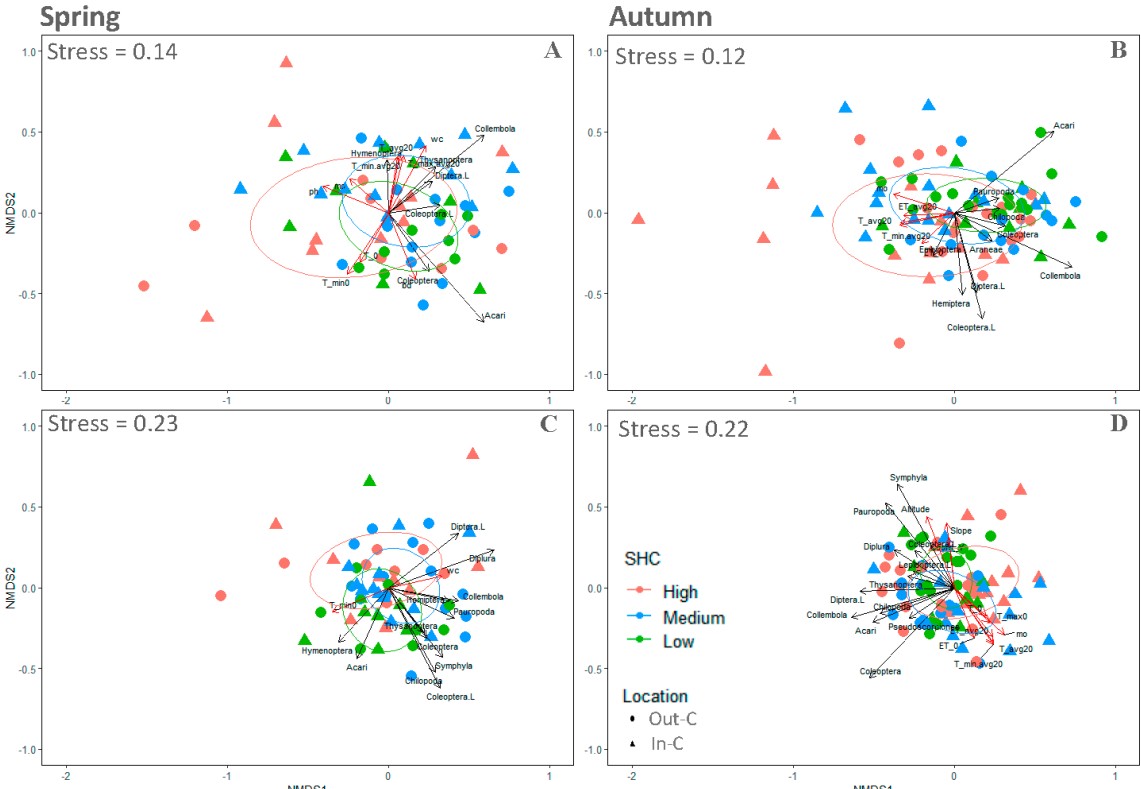

**Figure 4.** Community composition fits categorical predictors (SHC and location factor), environmental parameters and taxa (abundances and EMIs). Log-transformed abundance matrices for spring and autumn samplings are shown in graphics (**A**,**B**), respectively; EMI-values matrices for spring and autumn samplings are shown in panels (**C**,**D**), respectively. Black arrows show the fitting of significant taxa, whereas red arrows show the fitting of significant environmental variables. Location indicates whether the sampling point was established beneath the tree canopy or outside the canopy; SHC indicates the characteristics of the surrounding environment, as well as the pressure of the livestock where sampling points were placed; bd = bulk density; ET_avg20 = average evapotranspiration of the 20 days prior to sampling; mo = soil organic matter content; T_0 = mean temperature of the sampling day; T_avg20 = average temperature of the 20 days prior to sampling; T_max0 = maximum temperature of the sampling day; T_min0 = minimum temperatures of the sampling day; T_min.avg20 = average minimum temperature of the 20 days prior to sampling; wc = soil water content.

As the NMDS ordinations based on EMI matrices were almost random in both seasons, a clear effect of SHC categories causing differences among sampling points was not determined. Nonetheless, segregation of communities by locations was quite evident in autumn as indicated by PERMANOVA ($p < 0.001$) in Table 3 and Figure 4D. Moreover, the number of significant taxa that fit the scaling based on EMI scores was higher when compared to taxa abundance ordination in both seasons. Once again, variables fitting communities was higher on NMDS than in PERMANOVA, but after the model

selection, soil water content was the only significant variable related to spring communities ($p < 0.05$). In contrast, the autumn model was related to the variation of slope ($p < 0.01$) and mean temperature of the 20 days prior to sampling day ($p < 0.01$), including the location factor. Despite this, and similarly to abundances, total variances explained by EMI spring and autumn models were low ($R^2$ = 8.4 and $R^2$ = 19.7%, respectively).

**Table 3.** PERMANOVA results for matrices of log-transformed abundances and eco-morphological index score matrices. Significant results for environmental parameters causing dissimilarity are marked with asterisk: (*) = $p < 0.05$; (**) = $p < 0.01$; (***) = $p < 0.001$. Location indicates if the sampling point was established beneath the tree canopy (In-C) or outside the canopy (Out-C); SHC indicates the characteristics of the surrounding environment and pressure of the livestock where sampling points were placed; EP (–20) = effective precipitation of the 20 days prior to the sampling day; T (-20) = average temperature of the 20 days prior to the sampling day.

| Community Matrix | Season | Source of Dissimilarity | Df | F | $R^2$ |
|---|---|---|---|---|---|
| Log-transformed Abundances | Spring | Location | 1 | 2.674 | 0.041 * |
| | | Water content | 1 | 3.444 | 0.052 ** |
| | | pH | 1 | 2.480 | 0.037 * |
| | | Soil $CO_2$ efflux | 1 | 2.263 | 0.034 * |
| | | Residuals | 54 | | 0.836 |
| Log-transformed Abundances | Autumn | Location | 1 | 6.217 | 0.062 *** |
| | | SHC | 2 | 2.184 | 0.044 * |
| | | Slope | 1 | 3.173 | 0.032 * |
| | | OM content | 1 | 3.530 | 0.035 ** |
| | | T (−20) | 1 | 2.720 | 0.027 * |
| | | EP (−20) | 1 | 2.156 | 0.022 * |
| | | Residuals | 78 | | 0.779 |
| EMIs | Spring | Location | 1 | 2.097 | 0.034 |
| | | Water content | 1 | 3.155 | 0.051 * |
| | | Residuals | 57 | | 0.916 |
| EMIs | Autumn | Location | 1 | 11.329 | 0.111 *** |
| | | Slope | 1 | 5.055 | 0.050 ** |
| | | T (−20) | 1 | 3.456 | 0.036 ** |
| | | Residuals | 82 | | 0.803 |

*4.3. Spatial and Temporal Patterns of Abundances and QBS-ar*

GAMs demonstrated how both response variables (total abundances and QBS-ar) changed in spring and autumn in relation to the spatial structure of environmental variables (Table 4 and Figure 5). In general, variance explained by the models was very high ($R^2$ = 87.4% and $R^2$ = 89.1% for abundances; $R^2$ = 85.5% and $R^2$ = 91.8% for QBS-ar in spring and autumn, respectively), the largest effect of which was the contribution of the random effect in NDMS coordinates (F = 21.750 and F = 23.880 for abundances in spring and autumn; F = 12.870 and F = 37.430 for QBS-ar in spring and autumn, respectively). Moreover, the smoothness of the models was conditioned by location factor, where out-C significantly explained a wider proportion of the variance over the unexplained (F = 7.920 and F = 8.923 in spring and autumn, respectively) rather than in-C. Also, effective precipitation of 20 days prior to sampling day resulted statistically significant in autumn (F = 2.539). Otherwise in autumn, QBS-ar scores followed similar spatial patterns to SOM content in autumn (F = –2.214). It is important to note that smoothing was better when several variables such as pH, SOM and bulk density (spring abundances), temperature (autumn abundances), and effective precipitation (autumn QBS-ar) were available. Despite their insignificant effects on smoothing, AIC obtained better scores when they were present.

Figure 5A,B shows the spatial patterns of smooth isolines for total abundances. The maximum order of magnitude in autumn ($e^4$) indicates a higher variation than in spring ($e^2$). Instead, smooth isolines for QBS-ar reached variations from 0 to 80 in both seasons (Figure 5C,D).

**Table 4.** GAMs results for community metrics in each season. Significant results are shown in bold. Location indicates where sampling stations were located: outside the canopy (Out-C) or beneath tree canopy (In-C); SHC-low/medium/high indicates the characteristics of environment and pressure of the livestock in which points were located; T (−20) = average temperature of the 20 days prior to sampling day; EP (−20) = effective cumulative precipitation of the 20 days prior to sampling day.

| Metrics | Season | Parameter | F | p | $R^2$ |
|---|---|---|---|---|---|
| Log-transformed Abundances | Spring | **s(NMDS1, NDMS2)** | **21.750** | **<0.001** | **0.874** |
| | | **Location-out-C** | **7.920** | **<0.001** | |
| | | Location-In-C | −0.948 | 0.348 | |
| | | $SHC_{low}$ | −1.705 | 0.095 | |
| | | $SHC_{medium}$ | −1.444 | 0.156 | |
| | | pH | −1.416 | 0.164 | |
| | | OM content | −1.529 | 0.134 | |
| | | Bulk density | −1.536 | 0.132 | |
| | Autumn | **s(NMDS1, NDMS2)** | **23.880** | **<0.001** | **0.891** |
| | | **Location-out-C** | **8.923** | **<0.001** | |
| | | Location-In-C | 1.742 | 0.087 | |
| | | $SHC_{low}$ | 1.899 | 0.062 | |
| | | $SHC_{medium}$ | 1.066 | 0.291 | |
| | | T (−20) | −1.605 | 0.114 | |
| | | **EP (−20)** | **2.539** | **0.014** | |
| QBS-ar | Spring | **s(NMDS1, NDMS2)** | **12.870** | **<0.001** | **0.855** |
| | | **Location-out-C** | **35.507** | **<0.001** | |
| | | Location-In-C | −0.873 | 0.389 | |
| | Autumn | **s(NMDS1, NDMS2)** | **37.430** | **<0.001** | **0.918** |
| | | **Location-out-C** | **15.892** | **<0.001** | |
| | | Location-In-C | 0.107 | 0.915 | |
| | | **OM content** | **−2.214** | **0.030** | |
| | | EP (−20) | −1.888 | 0.063 | |

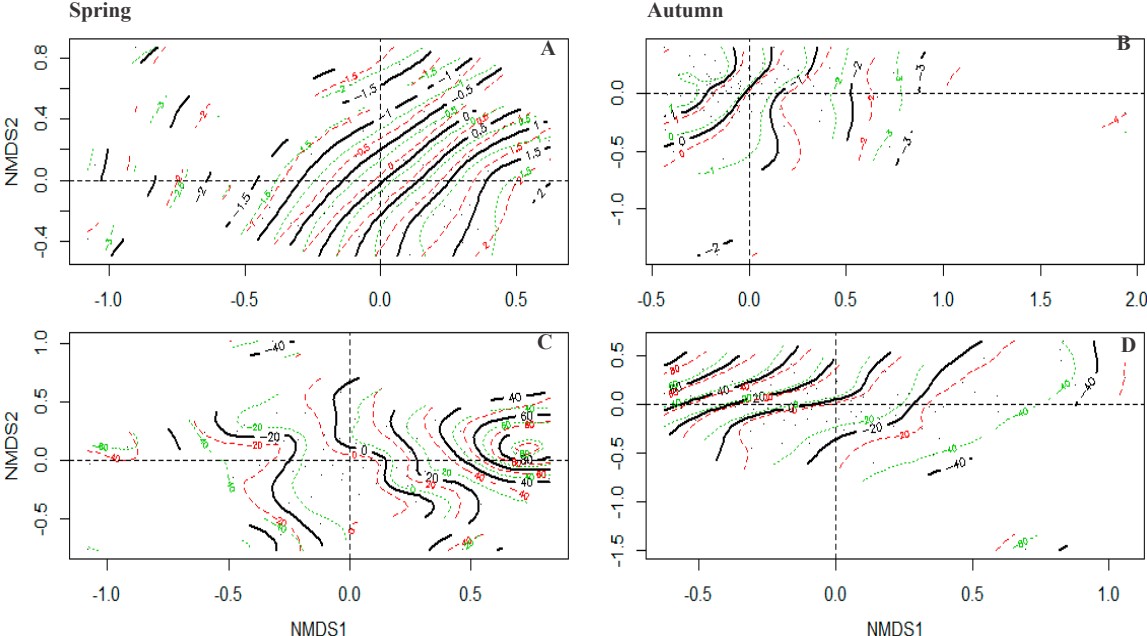

**Figure 5.** GAM plots representing spatial smoothing of the response variables. (**A**) GAM model for spring total abundances; (**B**) GAM model for autumn total abundances; (**C**) GAM model for spring QBS-ar; (**D**) GAM model for autumn QBS-ar. Codes represent the quantity by which each response variable varies. Black-solid isolines represent the spatial smoothing that belonged to a determined interval of variation. Red-dashed isolines represent the upper variation of that interval associated with each black solid isoline sharing the same code. Green-dashed isolines represent the lower variation of that interval associated to each black solid isoline sharing the same code.

## 5. Discussion

Understanding the ecosystem processes governing reservoirs of soil biodiversity, and the practices threatening them (e.g., anthropogenic mismanagement), would strongly benefit from characterizing the microarthropod community composition associated with traditionally managed rangelands. Moreover, the use of morphological traits in identifying spatial patterns and diversity of biological forms and relationships with above- and belowground environmental characteristics is a helpful method to detect areas under risk in terms of loss of soil multifunctionality [4,12]. This need becomes even more urgent since Mediterranean areas are especially sensitive to new climate change scenarios [7]. Features of Iberian rangeland ecosystems, such as the patchy distribution of vegetation and the unequal pressure exerted by livestock, may be a major structuring force of soil microarthropod communities at local scales. Nevertheless, several stochastic events (e.g., colonization and extinction) usually more related to regional scales can also take place locally [49].

### 5.1. The Response of Community Abundances

Our analyses suggest that abundances of microarthropod communities differed in response to the presence of trees and livestock pressure. These differences among community and taxa abundances can be simply explained by the environmental characteristics of each sampling point [50]. In this context, the analysis based on NMDS allowed us to identify dissimilarities among communities' compositions, and sensitivity of taxa populations in relation to spatial and temporal dynamics of niche-environmental parameters. Our results suggest that, in general, areas outside the influence of trees harbor higher abundance of microarthropods rather than areas beneath tree canopies. NMDS also reveals a great degree of overlap between locations × SHC categories that could be representative of a geographic dispersal of taxa (at the study area scale) due to the absence of physical barriers and proximity of categories [51]. It is noteworthy that approximately 80% of variation was undetermined by the multivariate analysis (PERMANOVA), thus, it could be due to other non-spatially structured environmental factors that were not measured in the field [52], or even because of biotic interactions within the microarthropod community due to its spatial aggregation [49,53]. Results of GAM, based on abundances in both seasons, support the results of PERMANOVA. GAM models reflected a clear spatial structure of total abundances, as indicated by approximately 88% of variance explained in both seasons. This indicates that spatiotemporal processes (e.g., dispersal) in relation to local environmental factors drive total abundances of microarthropod communities [52,54–56]. Therefore, soil environmental variables, characteristics of vegetation cover, and livestock pressure alone could not explain by themselves the community aggregation phenomena but a dispersion of total abundances. This fact coincides with [53,54], who concluded that the spatial structure of variables plus the spatial and temporal structure of total abundances reveals that overall abundances are mediated both by dispersal and environmental factors, where the effects of the latter are weaker.

Several relationships inferred from NMDS analysis are relevant. Significant taxa, such as Acari, Collembola, Pauropoda, Araneae, and Chilopoda, were related to out-C-SHC$_{medium}$ and out-C-SHC$_{low}$ in autumn, while Acari, Collembola, and Thysanoptera were positively associated with out-C-SHC$_{medium}$ and out-C-SHC$_{low}$ in spring. Such convergence of positive correlations in the same direction could indicate that lower livestock pressure allows the development of populations and the co-occurrence of microarthropod taxa in the absence of trees, which accords with Mulder et al. [57]. However, an ensemble of biotic and abiotic interactions can also cause such relationships: for instance, soil parameters analyses indicate that trees contribute to rising SOM values, mostly due to litter inputs [21], which provide habitat and energy budget to the detrital community. In addition, significant negative relationships between pH, SOM, and abundances of such groups as Acari and Coleoptera in spring, as well as between SOM, and Collembola and Chilopoda (and Coleoptera) in autumn, have been identified. Except for Chilopoda (mainly predators), the other taxa show a wide variety of feeding habits. Indeed, the spectrum of feeding habits within the microarthropod community is wide [5]. However, our study is limited since the assemblage of communities was performed at a low

taxonomical level. Thus, in order to shed light on relationships between feeding habits, environmental parameters, and community structure, an analysis of the functional roles of microarthropods at higher taxonomical level should be performed.

Another inference suggested by results based on NMDS analysis is that Acari and Collembola abundances avoid each other, as mites were more related to out-C, whereas collembolans were more related to in-C areas. However, this affirmation must be put in context: mites and collembolans are dominant and normally coexist in soil. Locally, competition for resources between both taxa (e.g., those with detritivorous feeding habits) is expected to benefit some functional groups over others due to negative interactions, as suggested by Caruso et al. [54] in a study of Antarctic microarthropod communities. As our study lacked a functional characterization of taxa, this affirmation is not sustained by our results. Otherwise, several authors [24,29] indicate that trees (and litter) in semiarid wood pastures decrease soil evapotranspiration rates by sunlight interception [31], and consequently decrease wider fluctuations of soil moisture when compared to open spaces. This causes a response of the entire microarthropod community to light availability as demonstrated by Jiménez-Chacón et al. [31], who concluded that detritivores preferred darker microsites (e.g., beneath tree canopies). Hence, a higher accumulation of litter and SOM beneath tree canopies, associated with lower rates of evapotranspiration, promoted collembolan abundances and likely inhibited the development of detritivorous mites by competition for resources. The opposite occurs in localities where mite populations are greater than collembolans, but our results do not allow for speculation about such dynamics.

### 5.2. The Response of Biological Forms' Evolutive Adaptation

In the context of this study, rangeland mismanagement leads to a promotion of undesirable vegetation in terms of livestock profit. However, from the perspective of biodiversity, a greater number of well-adapted microarthropod communities live in such areas. The measurement of biological forms' adaptation to soil environment involves the evaluation of such traits as depigmentation; reduction of appendages, such as antennae and legs; presence, absence, or reduction of the visual apparatus; presence, absence, or reduction of wings; dimensions; and body shape [11]. The higher the score attributed to each trait, the better adapted to a soil environment the organism is. The sums of all these scores serves as the EMI of each biological form. In the same way, the greater the number of biological forms with high scores that lives in the soil, the better quality and stability of the soil (i.e., the less disturbance) [58]. This is the main concept upon which QBS-ar relies and, by definition, it is based on environmental filtering theory. Overall, our analyses suggest that microarthropod communities' evolutive adaptation to their soil environment differed mostly in response to the presence of trees. These differences reinforce the hypothesis that vegetation cover and environmental characteristics (i.e., habitat degradation caused by livestock pressure) are major forces that structure microarthropod communities even at evolutive level, which accords with the main basis of QBS-ar and environmental filtering theory.

Stress values of the NMDS analysis based on EMI communities suggest that the goodness of fit on morphological traits was poor. Nevertheless, a clear and significant differentiation between out-C and in-C was identified (stronger in autumn than in spring). This supports our hypothesis that vegetation cover shapes the microarthropod adaptation, but it contradicts the result obtained by Meloni et al. [39], who studied the community composition of ground arthropods, in terms of abundances and richness, in patches of vegetation and interpatch in Mediterranean drylands. In that study, patches of vegetation harbored higher richness and abundances than in interpatch areas when compared to the open space values of our study. However, in Meloni et al.'s study it is noteworthy that vegetation cover on interpatch areas was absent (bare soil).

NMDS analysis also showed a great degree of overlap between SHC categories, and insignificant effects on EMI communities. Overall, a greater number of taxa showed significant fits to SHC areas and locations. Unfortunately, it is difficult to extract clear inferences in relation to SHC based on these results due to stress values over 0.20. However, disturbances driven by livestock could explain it

since several authors consider that well-managed silvopastoral systems, for instance with livestock charges at 1 AU ha$^{-1}$ or below [30], could enhance resource allocation within soil food webs [59], by, for example, altering the C:N ratio [60]. This fact supports the niche-environmental hypothesis, and it could explain why SHC$_{medium}$ showed similar patterns to SHC$_{low}$ on QBS-ar values.

Finally, poor values of variance explanation resulted from PERMANOVA. Approximately 87% of the total variance remained undetermined. This implies that the occurrence of morphological adaptation may be related to spatially structured variables or biotic interactions (or both) that were not measured in the field. Results of GAM analyses based on QBS-ar confirm the hypothesis that morphological adaptation also follows a spatially structured distribution. Moreover, it is even stronger than total abundances models. Smoothing patterns also differed from spring to autumn, which were negatively related to the spatial position of trees and SOM in autumn, and only to trees in spring. Therefore, the response to the third question of this work is that morphological adaptation and abundances did not follow identical, but similar, spatial patterns as confirmed by NMDS, PERMANOVA and, finally, GAM analysis.

*5.3. Object-Based Image Analysis and SHC Classification*

Correlations between microarthropod communities' structures, metrics and evolutive adaptations and SHC classification using OBIA were not as high as expected. This might have been due to the fine scale at which microarthropod populations develop themselves, or to the relatively rapid dynamics of annual grasses. OBIA was chosen as the best candidate to remotely classify objects on the ground when compared to pure pixel classification techniques, but it obviously presents problems regarding the pixel's dimensions of the image. However, OBIA turned out to be a useful technique to identify livestock effects, which would be a useful analysis when performed at larger scales. That being said, we were able to confirm that the results of the analysis corresponded with the areas in which livestock spend more time, as demonstrated in precedent studies about physical-chemical indicators of soil quality [25], impacts of livestock [30], and soil erosion studies [30,61] realized within the study area.

## 6. Conclusions

Results of this study suggest that there is a clear effect of spatial heterogeneity and spatially distributed variables (measured and unmeasured in the field) on structuring community metrics and community composition. This study demonstrates several facts: (1) landscape characteristics play a crucial role on the occurrence of evolutive adaptation of microarthropod biological forms; (2) abundances and the occurrence of morphological adaptation did not follow identical, but similar spatial patterns; (3) and the effect of environmental characteristics, such as the patchy distribution of vegetation being high on abundances and taxa diversity, which is likely due to environmental filtering, but to stochastic dispersal as well. In contrast, environmental filtering better explains the spatial distribution of QBS-ar and community composition based on EMI scores. (4) Higher abundances and adaptation to soil environment were related to open spaces rather than areas under arboreal influence. Smoothing of GAM models responded to the spatial positions of trees in terms of overall abundances and QBS-ar in both seasons. Moreover, the contribution to GAM models of the spatial structure of soil parameters and livestock pressure was unexpectedly low in abundances, and almost absent in QBS-ar patterns (with the exception of SOM content in autumn). This indicates that stochastic dispersal in relation to local environmental factors (e.g., non-spatially structured abiotic factors, as well as biotic interactions) drive abundances and adaptation of microarthropod communities. (5) High livestock pressure influenced microarthropod communities' composition and metrics. Better values of community metrics were reached in medium and low areas, which indicates that lower livestock activity trends to enhance microarthropod zoocenoses. (6) The delimitation of SHC areas via OBIA technique showed unexpectedly lower correlations with microarthropod communities' composition. For future studies, we strongly recommend the use of UAVs and multispectral images in order to reduce pixel dimensions, but also to determine the physiological state of vegetation. In this context,

identifying the links between vegetation and belowground communities might be crucial to accurately quantify the resilience of ecosystems, and the consequences of climate change for humankind.

**Author Contributions:** Conceptualization, development of the idea, as well as experimental design and sampling: C.L.F., J.B.G., and M.P.F.; laboratory activities: C.L.F. and J.B.G.; data provision for completion of the database: J.L.-P.; statistical analyses: C.L.F. and S.R.; writing—original draft preparation: C.L.F.; supervision of the whole work: C.M.; review and editing: C.M. and J.L.-P. All authors have read and agreed to the published version of the manuscript.

**Funding:** This research received no external funding.

**Acknowledgments:** We are very grateful to the GeoEnvironmental Research Group (GIGA) from the University of Extremadura who provide us with technical support, instruments, and facilities to carry out our analysis. Moreover, we are also grateful to undergraduate students who help us in laboratory works.

**Conflicts of Interest:** The authors declare no conflict of interest.

## Appendix A

**Table A1.** Means ± SD of meteorological parameters calculated for the period 01/01/2013 to 31/12/2018. From: Redarex; Meteorological station: Valdesalor (CC18) which, is approximately 31 km away from the study area; altitude: 382 m.; coordinates UTM H30 X: 730,101, Y: 4,361,000; Extracted from [62].

| Meteorological Variable | Units | Value | | |
|---|---|---|---|---|
| Annual solar radiation | W/m$^2$ | 16.59 | ± | 6.76 |
| Net solar radiation | W/m$^2$ | 7.52 | ± | 4.22 |
| Mean annual temperature | °C | 15.02 | ± | 6.30 |
| Maximum mean temperature of the coldest month | °C | 15.84 | ± | 0.62 |
| Minimum mean temperature of the coldest month | °C | 2.41 | ± | 0.27 |
| Maximum mean temperature of the warmest month | °C | 30.60 | ± | 6.46 |
| Minimum mean temperature of the warmest month | °C | 13.61 | ± | 4.22 |
| Mean annual rainfall | mm | 524.2 | ± | 28.4 |
| Mean annual effective precipitation | mm | 249.8 | ± | 14.4 |
| Mean annual evapotranspiration | mm | 1363.1 | ± | 75.5 |

**Table A2.** Eco-morphological Indexes (EMIs) score for each microarthropod taxa. Groups shown are those found in this work.

| Taxa | EMI Score |
|---|---|
| Pseudoscorpiones | 20 |
| Opiliones | 10 |
| Araneae | 1–5 |
| Acari | 20 |
| Isopoda | 10 |
| Diplopoda | 10–20 |
| Pauropoda | 20 |
| Symphyla | 20 |
| Chilopoda | 10–20 |
| Protura | 20 |
| Diplura | 20 |
| Collembola | 1–20 |
| Psocoptera | 1 |
| Hemiptera | 1 |
| Thysanoptera | 1 |
| Zigentomi | 10 |
| Embioptera | 10 |
| Orthroptera | 1–20 |
| Coleoptera | 1–20 |
| Hymenoptera | 1–5 |
| Diptera | 1 |
| Lepidoptera | 1 |
| Coleoptera (larvae) | 10 |
| Diptera (larvae) | 10 |
| Hymenoptera (larvae) | 10 |
| Lepidoptera (larvae) | 10 |
| Holometabolans (adults) | 1 |

**Table A3.** Significant comparisons from post hoc Tukey tests are shown.

| Metrics | Factors | Pairs Comparison | Difference | p |
|---|---|---|---|---|
| Ln abundances | Livestock pressure | Low − High | 0.568 | 0.045 |
| | Location | In-C − Out-C | −0.07 | <0.001 |
| | Season | Spring − Autumn | −0.051 | 0.007 |
| | Livestock pressure × Location | High × In-C − Low × Out-C | −1.108 | 0.009 |
| | | High × In-C − Medium × Out-C | −1.147 | 0.004 |
| | | Medium × In-C − Medium × Out-C | −0.9 | 0.03 |
| | Livestock pressure × Season | Low × Autumn − High × Autumn | 0.882 | 0.038 |
| | | High × Spring − Low × Autumn | −1.093 | 0.032 |
| | | Medium × Spring − Low × Autumn | −0.924 | 0.036 |
| | Location × Season | In-C × Autumn − Out-C × Autumn | −0.651 | 0.036 |
| | | In-C × Spring − Out-C × Autumn | −1.199 | <0.001 |
| | Livestock pressure × Location × Season | High × In-C × Spring − Low × Out-C × Autumn | −1.526 | 0.048 |
| | | High × In-C × Autumn − Medium × Out-C × Autumn | −1.427 | 0.043 |
| | | High × In-C × Spring − Medium × Out-C × Autumn | −1.733 | 0.019 |
| | | High × In-C × Spring − Low × In-C × Autumn | −1.781 | 0.035 |
| QBS-ar | Location | In-C − Out-C | −19.693 | <0.001 |
| | Livestock pressure × Location | High × In-C − High × Out-C | −24.222 | 0.037 |
| | | High × In-C − Low × Out-C | −28.101 | 0.009 |
| | | High × In-C − Medium × Out-C | −33.791 | <0.001 |
| | | Medium × In-C − Medium × Out-C | −23.922 | 0.027 |
| | Livestock pressure × Season | High × Spring − Low × Autumn | −23.295 | 0.09 |
| | Location × Season | In-C × Autumn − Out-C × Autumn | −27.661 | <0.001 |
| | | In-C × Spring − Out-C × Autumn | −24.243 | 0.002 |
| | Livestock pressure × Location × Season | High × In-C × Autumn − High × Out-C × Autumn | −39.419 | 0.08 |
| | | High × In-C × Autumn − Low × Out-C × Autumn | −34.004 | 0.051 |
| H′ | Location × Season | In-C × Spring − Out-C × Spring | 0.197 | 0.063 |

**Table A4.** Average QBS-ar values ± SD found at each SHC (high, medium and low livestock pressure), in each Location (beneath = in-C and outside the canopy = out-C) during both sampling campaigns.

| | Spring | | Autumn | |
|---|---|---|---|---|
| Livestock Pressure | Out-C | In-C | Out-C | In-C |
| $SHC_{high}$ | 66.9 ± 28.3 | 72.1 ± 27.2 | 104.4 ± 34.2 | 64.9 ± 23.2 |
| $SHC_{medium}$ | 97.1 ± 39.4 | 80.3 ± 22.2 | 105.9 ± 27.8 | 76.8 ± 32.3 |
| $SHC_{low}$ | 91.3 ± 22.4 | 84.8 ± 22.7 | 98.9 ± 18.4 | 89.0 ± 31.5 |

**Table A5.** Average Shannon's diversity values ± SD found at each SHC (high, medium and low livestock pressure), in each Location (beneath = in-C and outside the canopy = out-C) during both sampling campaigns.

| | Spring | | Autumn | |
|---|---|---|---|---|
| Livestock Pressure | Out-C | In-C | Out-C | In-C |
| $SHC_{high}$ | 0.79 ± 0.36 | 0.78 ± 0.33 | 0.97 ± 0.27 | 0.89 ± 0.23 |
| $SHC_{medium}$ | 0.76 ± 0.20 | 1.04 ± 0.41 | 0.86 ± 0.34 | 0.83 ± 0.29 |
| $SHC_{low}$ | 0.69 ± 0.29 | 1.00 ± 0.48 | 0.76 ± 0.26 | 0.68 ± 0.24 |

**Table A6.** Absolute numbers of microarthropod found at each SHC (high, medium and low livestock pressure), in each Location (beneath tree canopy = In-C or outside the canopy = Out-C) during the first sampling campaign.

| | Spring | | | | | |
|---|---|---|---|---|---|---|
| | Out-C | | | In-C | | |
| Taxa | $SHC_{high}$ | $SHC_{medium}$ | $SHC_{low}$ | $SHC_{high}$ | $SHC_{medium}$ | $SHC_{low}$ |
| Pseudoscorpiones | - | 3 | - | - | - | 3 |
| Opiliones | - | - | - | - | - | - |
| Araneae | 1 | - | 3 | - | 2 | 2 |
| Acari | 3410 | 3928 | 2882 | 902 | 1116 | 2030 |
| Isopoda | - | - | - | - | - | - |
| Diplopoda | - | - | - | - | - | 2 |
| Pauropoda | 1 | 8 | 1 | 3 | - | - |
| Symphyla | - | 10 | 3 | - | 3 | - |
| Chilopoda | 2 | 17 | 18 | 4 | 19 | 8 |
| Protura | - | 1 | - | - | - | 3 |
| Diplura | 30 | 29 | 24 | 3 | 7 | - |
| Collembola | 995 | 4368 | 1046 | 3424 | 5584 | 936 |
| Psocoptera | - | - | - | 5 | 2 | 7 |
| Hemiptera | 1 | 51 | 30 | 14 | 34 | 13 |
| Thysanoptera | 1 | 24 | 7 | 6 | 20 | 8 |
| Zigentomi | - | - | - | - | - | - |
| Embioptera | 1 | 2 | - | 6 | 4 | 2 |
| Orthroptera | - | - | - | 1 | - | - |
| Coleoptera | 46 | 180 | 28 | 25 | 18 | 12 |
| Hymenoptera | 5 | 26 | 64 | 17 | 153 | 31 |
| Diptera | - | 1 | 2 | 2 | 7 | 6 |
| Lepidoptera | - | - | 1 | 1 | 1 | - |
| Coleoptera (larvae) | 36 | 31 | 30 | 14 | 31 | 34 |
| Diptera (larvae) | 29 | 218 | 63 | 64 | 120 | 135 |
| Hymenoptera (larvae) | - | - | - | 4 | - | - |
| Lepidoptera (larvae) | 1 | 3 | 1 | - | 2 | 2 |
| Holometabolans | - | - | - | - | - | - |
| Total | 4559 | 8900 | 4203 | 4495 | 7123 | 3234 |

**Table A7.** Absolute numbers of microarthropods found at each SHC (high, medium and low livestock pressure), in each Location (beneath tree canopy = In-C or outside the canopy = Out-C) during the second sampling campaign.

| | Autumn | | | | | |
|---|---|---|---|---|---|---|
| | Out-C | | | In-C | | |
| Taxa | $SHC_{high}$ | $SHC_{medium}$ | $SHC_{low}$ | $SHC_{high}$ | $SHC_{medium}$ | $SHC_{low}$ |
| Pseudoscorpiones | 1 | 6 | 1 | - | 1 | - |
| Opiliones | - | - | - | - | - | - |
| Araneae | 1 | 5 | 12 | 4 | 3 | 1 |
| Acari | 4003 | 5375 | 10370 | 1321 | 2599 | 2498 |
| Isopoda | - | - | - | - | 1 | - |
| Diplopoda | - | - | - | - | - | - |
| Pauropoda | 16 | 383 | 56 | 1 | 29 | 97 |
| Symphyla | 18 | 9 | 20 | 7 | 1 | 2 |
| Chilopoda | 5 | 11 | 2 | - | 11 | 1 |
| Protura | 2 | 2 | 7 | - | 1 | 1 |
| Diplura | 8 | 1 | - | - | 3 | 1 |
| Collembola | 7079 | 9787 | 13472 | 4708 | 4892 | 10,574 |
| Psocoptera | - | - | - | - | - | 2 |
| Hemiptera | 102 | 22 | 18 | 243 | 12 | 17 |
| Thysanoptera | 34 | 29 | 11 | 7 | 36 | 13 |
| Zigentomi | - | - | 3 | 1 | - | - |
| Embioptera | - | - | - | 7 | 2 | 5 |
| Orthroptera | - | - | - | - | - | - |
| Coleoptera | 96 | 85 | 29 | 10 | 36 | 32 |
| Hymenoptera | 3 | 17 | 17 | 80 | 58 | 10 |
| Diptera | 31 | 15 | 16 | 7 | 25 | 13 |
| Lepidoptera | - | - | - | - | - | - |
| Coleoptera (larvae) | 152 | 101 | 92 | 137 | 52 | 69 |
| Diptera (larvae) | 1374 | 712 | 226 | 238 | 84 | 71 |
| Hymenoptera (larvae) | - | - | - | - | - | - |
| Lepidoptera (larvae) | 70 | 16 | 21 | 21 | 4 | 2 |
| Holometabolans | 3 | 6 | 1 | 1 | 2 | 1 |
| Total | 12,424 | 16,382 | 24,310 | 6700 | 7839 | 13,410 |

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
