# Peer review of "Effects of Livestock Pressure and Vegetation Cover on the Spatial and Temporal Structure of Soil Microarthropod Communities in Iberian Rangelands"

_forests, doi:10.3390/f11060628_

Round 1

Reviewer 1 Report

Line 18: please replace “perform a threat” with “pose a threat”;

Line 32: if the “photo interpretation analysis” was mentioned previously in the abstract, it seems to be a surplus information on the same method in the text that should as brief as possible;

Lines 32 – 33: “showed a slightly or slight trend?”

Line 38: abbreviations such as QBS-ar and OBIA used as keywords – do authors think that readers are familiar with that enough?

Line 52: “they are a key component”;

Lines 92 – 94: there is used twice “agro-silvo-pastoral”;

Line 96: “composed of annual species”;

Line 103: please reconsider the order of words “study area is in old erosion surfaces formed”;

Line 104: please remove the last “s” in “soils are shallows”;

Line 116: please add such to “parameters such as”;

Lines 124 and 126 - 127: Retama sphaerocarpa L. should be Retama sphaerocarpa L.;

Lines 143 – 144: T and G seemed to me to be the principal treatments compared – why did authors not mention them in the abstract first?

Line 145: “stablished” – is it a common verb?

Line 146: “in order to obtain the best soil community” – is not this approach posing a threat of bias?

Line 149: does the “station” mean the same as the sampling point?

Line 151: “executed” or “conducted”? Which is better?

Line 153: “leveraged” - term from economy?

Line 161: “Loss of Ignition” should be “loss-on-ignition”?

Line 168: 40×

Lines 180 – 181: please remove “each” and add plural – i.e. “test was performed to evidence statistical differences among categories

Line 183: “doubt to” used for the first time: What does it mean? When looking for the "doubt to" in the internet, Google Scholar returned more than 2.9 million results containing "doubt" but no result with "to" included. Is that something similar to "owing to"? It should be revised by native speakers of English.

Table 1: “SHC low” values are not presented appropriately!

Figure 3: The same and/or similar colors of horizontal lines make the picture "unfriendly" to readers.

Lines 282 – 283: “inductors of dissimilarity” – is it appropriate?

Line 306: begin sentence with (A and B)?

Line 323: “were founded” or “were found”?

Figure 5 is captioned “GAM plots” – the same as GAMM?

Line 357: “demolition of organic matter” or “decomposition”?

Lines 367 – 368: “disturbs associated to” or “disturbances associated with”?

Lines 369 – 371: Indeed, variations in taxonomical composition and functional groups (or even, changes in nutrient dynamics) could lead to loss of taxa and compromise the ecosystem services relying on them [15]. I do not understand the meaning.

Line 376: „closer among them” or “closer to each other”?

Line 377: Some authors (Citations missing) consider ...

Line 389: spend more time or what?

Line 403: “were founded” or “were found”?

Line 411: please, delete “s” in “precipitations” – as for the rain, snow etc., it is always precipitation.

Line 416: Google Scholar returned more than 7 million results after entering words "with also". Unfortunately, I did not find any example of their use as such.

Line 419 - 420: please reconsider the wording of “Despite this, highly adapted to soil environment biological forms appeared only in stations located where the impact of livestock pressure was lower”.

Line 425: please add “of” to “decrease of the diversification

Line 429: soil disturbances?

As for the whole manuscript, authors begin the sentences very often with words such as “moreover, however, indeed, therefore, thereby, although, nevertheless, furthemore”. Please, try to find out the appropriate frequency of these important words in texts written by those whose mothers tongue is English. I think you will see these are something like “seasoning” that help understand better but should not be overused.

There are also many abbreviations such as QBS, OBIA, GAMM, NMDS1, NDMS2 etc. I have seen many studies where abbreviations are used and also I have used them. However, if they are too many, it makes troubles for readers to get through.

Reviewer 2 Report

The manuscript forests-771008 “Effects of pasture and vegetation cover on the structure of soil microarthropod communities in Iberian rangelands” is consistent with the objectives of the journal and an analysis of the response of soil animals could be useful in understanding mechanisms governing the dynamics of microarthropods communities in particular degraded soils.

Some parts of the elaborate are not clear and oversimplifications in the use of bibliographic informations are evident both in the Introduction and in the Discussion sections.

One of the aims of the manuscript is worthy of note, because investigating changes in the spatial and temporal structure of edaphic arthropod communities, depending on the environmental characteristics, is really interesting. However the work, written and organized in a way that I would call superficial, fails to shed light on these issues. Nowhere in the manuscript particular morphological adaptations related to the diversity of the various environments investigated have been described. The only morphological adaptations that can be interpreted are those characterizing biological forms used in the attribution of EMI (eco-morphological index) and therefore necessary for the calculation of the QBS-ar index. The authors should be aware that the QBS-ar index was not designed considering functional groups but biological forms (biological forms don’t necessarily include functional groups) which are based on the sharing, by evolutionary convergence, of morphological adaptations to the soil environment. On these basis it may be possible to indicate which taxa are more or less sensitive to environmental stress and its variability in time and space. In this sense, it could have been useful to add a table with QBS-ar values in the Appendix.

Although in the introduction of the manuscript the authors discuss micro-arthropod communities, reading the work, problems related to an incomplete/superficial analysis of obtained results seem to emerge. This fact is highlighted by the lack of real and convincing discussion. In fact, in this section the authors have not been able to intercept real links between data and results obtained, since I believe that, although determined at a high taxonomic level, zoocenoses had to be evaluated in another way avoiding trivial and speculative conjectures with other works cited in the references that sometimes didn't match with obtained results. Although the authors have used a good level bibliographic material, they have not been able to use it wisely in organizing the discussion.

It is really difficult to read and understand the whole discussion. Please rewrite this part by correcting the construction and grammar of the periods, improving english language.

In any case, the authors through extensive and well-organized sampling strategy and utilizing elaborate statistical analysis have obtained results that can and must be described with greater attention and formal correctness, avoiding trivialities and speculations seen in the interpretation and discussion of the results.

So I feel that it should be redrafted and resubmitted after major revision.

Remarks and weaknesses:

  1. Introduction

 Lines 58–60. “Many ecological functions have been attributed to soil microarthropod communities. However, functions they accomplish can be compromised because of the reduction of diversity in functional species and their abundances”. – Although citation [14] talked about relationship between plants and three functional groups of nematods (which are really different with respect to soil microarthropod fauna in their role and function), what’s the meaning of “functional species” (at least among microarthropods) and why their reduction could be fundamental in maintaining soil health?

Line 63. “…and even feature the subentry of processes as organic matter photochemical degradation [16]”. Not clear. what’s the meaning of this statement? Looking at cited paper [16], the research deals with extreme drought events able to cause dramatic changes in ecosystem structure and function. Whole ecosystem CO2 fluxes, leaching losses, Litter decomposition, C/N stocks in vegetation and soil microbes were recorded. Results suggest that plant–soil interactions play a key role for the short-term stability of above-ground vegetation to extreme drought events.  

Lines 73-76. “However, many authors have remarked trees effect in dehesas as “island of biodiversity” because of the role they play regulating environmental features such as soil temperature and moisture and enhancing niche-environmental characteristics such as availability of nutrients ”. What’s the meaning of “island of biodiversity”. Cited paper [24] talks about vascular plants, earthworms, bees and spiders in Dehesas. Although this information could be useful it is not completely exaustive of such statement. Only earthworms are the only animals really related to soil and in this case they show low diversity in wood pastures. The great diversity cited in the paper refers to the great habitat mosaic with numerous marginal habitats present in this part of Spain. No reference to soil arthropods has been made in the other citations ([25] [26]) present in the Introduction section where we can read papers dealing with fine roots and mycorrhizas or how interactions between soil moisture and vegetation covers influence soil temperatures in very water-limited environments.  In these papers indications of the heavy-grazing effects, produced  by the excessive number of animals, on soil quality and pasture production are given. No reference to soil fauna has been reported. It is possible to find references in literature on microrthropods?

  1. Study area

Figure 1. I would modify this image by trying to refer the investigated area to the entire Spanish territory. In this sense a new Figure 1 could be used both in paragraph 2 (Study area) and paragraph 3.1 (Determination of the intensity of livestock pressure using photointerpretation)

Probably Table S2 in Appendix, as represented in this form, could be consider redundant. These information are not functional to the elaborate.

  1. Material and methods

3.1. Determination of the intensity of livestock pressure using photointerpretation

Lines 152-157. “the soil volume inside the cylinder used to determine the soil CO2 efflux was leveraged as a sample itself and took to the laboratory, where pH, soil organic matter content (SOM) analyses and microarthropod extraction were carried out in order to characterize the microhabitat of each station. In parallel, 3 measures of soil moisture were taken in 3 points around the cylinder using a TDR device; moreover, 3 undisturbed soil cores were collected using a steel cylinder at knew volume (100 cm³) and a soil sample extractor. Described sampling tecnique is unclear. In particular it is not clear what kind of soil samples have been used for faunal sampling. Are the authors able to quantify the volume and deep of the collected soils? This is a very important information when microarthropods communities are investigated.

3.3. Microarthropod extraction.

The informations provided in this paragraph are insufficient. I don't understand why the authors mention only Hexapoda, Crustacea, Chelicerata and Myriapoda (although this taxonomic rank has been suppressed for years and currently only Diplopoda, Chilopoda, Pauropoda and Symphyla classes are recognized) if, reading Tables S4 and S5, it seems that they have fully and correctly applied the QBS-ar index on entire arthropod fauna. A brief description of QBS-ar index is required. This may be helpful in understanding both general idea of the manuscript and data used in statistical analysis. After these considerations, I recommend the authors to change the title of the paragraph. I propose "analysis of the microartropod community". Or something similar.

3.4. Statistical analyses

Lines 183-185. “Once source of variation doubt to seasonal category was statistically confirmed, the following tests were applied considering both spring and autumnal campaigns differently”. Not clear, please write in better english. What’s the meaning od “doubt”? This verb occurs throughout the manuscript

  1. Results

4.3. Response of the community metrics to seasonality, vegetation cover and livestock pressure

Line 229. “Generally, 113579 organisms belonging to 27 taxa were singularly identified and counted”. Here authors report number of individual but in Table S4 and Table S5 they provide abundances/m2. I know that sometime this kind of information is given even if I am still not able to understand why. Please let try to make uniform the data.

Line 243. (F=14.655 for abundances and F=17.464 for QBS-ar, but not significant for H’).” Redundant. These results are present in Table 2

Line 244-245. Indeed, much lower abundances of proturans were detected in T (285 individuals m2, respectively), when compared to those abundances detected in G (≈1300 individuals m-2). At this point, a doubt arises: did the authors use the crude abundances or they utilized abundances per square meter?

Lines 245-248. “Although, SHC was not revealed as significant source of variation by the ANOVA on the community metrics, some taxa were specific from SHClow when trees were present such as isopods, diplopods or embiopterans, which abundances were higher when compared to SHCmedium and SHChigh, or even lacked”. Reading Tables S4-S5, Isopods seem really specific when trees are present but in SHCmed not in SHClow. Diplopods are only present only in autumn in SHClow G and in SHClow T. Embiopteran, In reality, seem widespread all over the areas in Spring while in autumn they have been collected only under Trees. How can the authors state that Embiopteran are specific in SHClow sites? And again, I am not able to understand the cited values of 1300 and 13000 individuals m2.

Lines 266-267. The effects of the tree influence area and livestock pressure were more evident on the community structure more than onto community metrics. – This concept is not clear. Please explain, even if I consider that this statement has to be treated in discussion section.

Lines 267-269. “According to results presented in Table S4 and Table S5 of the appendix A, the community structure varied as the number and demography of taxonomic groups did.” It is really trivial. It is obvious that community structures vary with species richness and demography. But are the authors sure it's correct to talk about demography? We all know that Demography encompasses the study of populations size, structure, distribution, spatial and temporal changes but in response to births, deaths, migrations, etc. It is impossible to talk about demography without these traits. Probably the authors should take into account the phenology of these arthropods which is rather well known.

Line 274. Please correct F value in text. In table 3 I read: F=2.674; R²= 0.041*

Line 278. “However, soil parameters slightly affected the structure of microarthropod communities”. Not clear. What is the meaning of “slightly”.

Lines 283-284. “Despite this, total variance explained by both the models was still low (R2=19.9 % in spring and R2=22.1 % in autumn)”. This evidence is really important. Why didn't the authors discuss this important result? Very often when we consider separately the effects dependent on the environmental and spatial variables we note that only a small percentage of community variation was explained while the remaining percentage of this variability remains unexplained and probably is related to spatial patterns independent of measured environmental variables. So this variation could be due to either unmeasured but spatially structured variables or to stochastic drift mediated by dispersal. Probably edaphic zoocoenoses are dominated by processes involving both deterministic and stochastic components and operating at multiple scales.

Lines 311-315. Results of GAMM models in text don’t match with table 4. For abundances in autumn R2 =89,1 not 85,5. For QBS-ar in spring R2 = 85,5 not 89,1. Please correct.

  1. Discussion

Lines 341-351. I am really not able to understand the introductory part of the discussion section. Please rewrite this part by correcting construction and grammar of the periods trying to avoid speculative and incomprehensible considerations.

A couple of examples below

Lines 345-346. “Although, not particularly well-developed microarthropod communities were detected in the whole study area, higher values of QBS-ar were correlated with the pasture layer”? What’s the meaning of: “not particularly well-developed microarthropod communities”? Are the authors sure that high QBS-ar values can be related to well diversified edaphic communities?

Lines 361-362. “However, our results inferred that vegetation, meaning microbial populations associated to the roots, had far more influence shaping microarthropod communities”. How can presented data be related to roots microbial populations?

  1. Conclusions

The same considerations made for the discussion section can be extended to the conclusion section.

Line 418-419. “…..but results were not clearly determinant for the entire community, but for several taxa doubt to the high percent of unexplained variation.” Non clear, please explain in better way.

Lines 420-421. “Furthermore, biotic interactions seemed to shape the community structure rather than soil abiotic parameters”. This deduction is purely speculative. Reduced biodiversity in soils may impair numerous ecosystem functions, such as nutrient acquisition by plants and the cycling of resources between above- and below-ground communities. So it is probably true but not supported by any direct data as confirmed in the following period by the authors themselves.

Ines 424-426. “In general, biological soil quality of this area was reduced and trees seem to exert an effect which contributes to decrease the diversification of microarthropod communities, probably doubt to the intensification of competition close to radicular systems”. This statement is highly speculative

Appendix

Table S4 and S5 don’t match. Please uniform the sites acronyms in each table.

Reviewer 3 Report

The paper deals with new information, useful on international level. The methods are explained clearly, presented in details. The results are based on statistically correct evidences. The list of references is rich enough. I propose minor revisions only:

38-39: in key words: please prefer singular, I mean niche instead of niches

42-47: for the initial part of Results: more cited works are needed (e.g. for the sentence „Soils are one of the most important reservoirs of biodiversity in the world“ etc.)

43: climatic change should be substituted by climate change

79-83 or 82-83: to add to the methods???

91: Could be moved to the first part of the Methods?

124: Retama sphaerocarpa should be written by Italics

173: Excellent statistics ...

225-227: bad table format

229: What does it mean taxa here? Please explain/define ... (it can be species, genus, family etc.)

309: An important and valuable sub-chaper!

338-339: Is there any chance to move these rows to the end previous page?

455-456: No clear table name (abundance of what etc.) and bad table size

455-456: The first taxonomic column has no name

458-459: No clear table name (abundance of what etc.) and bad table size

458-459: The first taxonomic column has no name

462-596: The list of references has not been checked (neither relations/connections with the text body nor the format according the rules of the editorial board)

Round 2

Reviewer 2 Report

The authors were able, after the notes provided by the reviewers, to create a completely different manuscript from the previous one. Now the work is organic, detailed and worthy of reading for all researchers interested in soil fauna.

Although there are only few small details to correct, I believe the work can be published in a high ranking magazine as "Forests"

Notes can be read below:

Figure 1. Please resize Figure 1 A (Image of Spain)

Figure 4. Please correct images paying attention to the contour lines of each ordination.

line 52-53. “they are key a compartment enhancing ……….”.  Plese correct in “they are a key compartment enhancing ………...

Lines 114-115. “The distribution of livestock charges and  equivalent units in 2018 are showed in appendix A (Table S2)” . This period represents a misprint of the old manuscript. Please remove from the present version. In the current version, table S2 indicates the eco-morphological indices and is correctly reported on line 184

Line 146. Change "Figure 1" with "Figure 1B"

Line 169 – “3.3. Analysis of the microarthropod community”. I apologize because I recommended this title but it is more correct to write: “Analysis of the microarthropod communities”

In caption of tables S6-S7 I still read “Average abundances of microarthropod per m²“ Please correct.
